# Glacial Debris Flow Blockage Event (2018) in the Sedongpu Basin of the Yarlung Zangbo River, China: Occurrence Factors and Its Implications

Baojuan Huai [1], Minghu Ding [2,*], Songtao Ai [3], Weijun Sun [1], Yetang Wang [1] and Jiajia Gao [4]

1   College of Geography and Environment, Shandong Normal University, Jinan 250014, China; 616070@sdnu.edu.cn (B.H.); 612033@sdnu.edu.cn (W.S.); 615002@sdnu.edu.cn (Y.W.)
2   State Key Laboratory of Severe Weather, Chinese Academy of Meteorological Sciences, Beijing 100081, China
3   Chinese Antarctic Centre of surveying and Mapping, Wuhan University, Wuhan 430079, China; ast@whu.edu.cn
4   Tibet Institute of Plateau Atmospheric and Environmental Sciences, Lhasa 850000, China; gaojj12@lzu.edu.cn
*   Correspondence: dingmh@cma.gov.cn

**Highlights:**

1. The topography and climate background determine that the hazard happens periodically, and these factors are fundamental for the weak vulnerability in the Sedongpu basin;
2. The dynamic glacier simulations with the Elmer/Ice model showed that the glacier surface velocity can reach 19 cm/d on the Dongpu glacier;
3. Heavy rain and an earthquake were triggering factors of the ice avalanche that led to the glacial debris flow.

**Abstract:** In this paper, the glacial debris flow blockage event, on 17 October 2018, in the Sedongpu basin of the Yarlung Zangbo River is taken as an example to analyse the occurrence and development of glacier hazards in this region. Multi-sources including remote sensing products, DEM, earthquake records and meteorological data were used to analyse the characteristics and mechanism of glacier hazards. The Elmer/Ice dynamic model was chosen to simulate the glacial surface velocity. It was found that topography and climate background determine that the hazard happens periodically. Based on the meteorological records of the Linzhi station, the warming rate was greater than 0.40 °C/10a during the period 1960–2017. The short-term heavy rainfall with daily values of 9.3 mm before the blockage event was also regarded as a factor. Both heavy rain and earthquake were triggering factors of the ice avalanche that led to the glacial debris flow. The glacier surface velocity of the Dongpu glacier simulated by Elmer/Ice model can reach 19 cm/d. This study has extensive applicability significance in glacier hazard mitigation under a changing climate.

**Keywords:** glacier hazards; ice avalanches; river blockage; climate change; Sedongpu basin; Yarlung Zangbo River

## 1. Introduction

Glacier hazards refer to hazards caused by their own instability, movement, and rapid changes [1], including glacial floods [2], glacial lake outbursts [3], glacial surge [4], glacial debris flow [5] and glacier collapse [6], etc., and mainly occur in high mountainous regions [7]. Glacier hazards receive quite a lot of attention because these hazards can directly cause great infrastructural damage and even cost lives [7]. There are also many indirect effects of glacial hazards on human society such as the adverse effects on mountain tourism, water supply shortages, and related socio-economic consequences [8]. Studies have shown that the Alps region has raised the most concerns, and followed by the Himalayas and the Qinghai-Tibetan Plateau (TP) [7].

As the "amplifier" of global warming, the TP has a more obvious response to climate change [4]. Regional climate change has raised the acceleration of glacier melting and thus increasing the risk of glacier hazards [7]. Studies have pointed out that the frequency and intensity of glacier hazards are increasing in TP and have caused a series of eco-environmental challenges and great economic loss [9]. For example, the Karayalak glacier surge in eastern Pamir on 4 May 2015 flooded 1000 hectares of grassland [10]; then, two glaciers near Aru Co collapsed on 17 July and 21 September 2016, killing nine people [11,12]. In 1950, a big earthquake in the TP, with a magnitude of 8.5, triggered a series of glacial debris flows [13]. These glacier hazard events have attracted much research interest (e.g., [14–16]).

The southeast TP is a region prone to landslide and other geological hazards, so it is also called the "Natural Disaster Museum" [1]. Cryospheric components, especially glaciers and snow cover supply a large amount of runoff to the rivers passing through and provide water for the local people [14,17,18]. The glacier shrinkage and hydrological responses occurring in the basins associated with the Yarlung Zangbo River (YZR) are remarkable and have attracted intense attention because these rivers provide water to large numbers of people, not only on the TP, but also in South Asia (e.g., [19–22]). There are many records of debris flow events [15,16]. For example, in the YZR Sedongpu basin, up to four large-scale river-blocking events induced by debris flow occurred in the 2018~2019 period [16].

Studies of the formation mechanism, occurrence process and damage related to glacial hazards are of great significance for mitigation and prevention [23]. Topography, precipitation and air temperature are considered as main factors initiating glacier hazards, plus others such as the sudden emptying of ice-marginal lakes and increase in pore water pressure [24]. The glacier's surface velocity is an indicator of glacier stability in the study of mountain glacier dynamics as a result of climate forcing [25]. One method is to solve partial differential equations for glacier dynamics and thermodynamics based on a function of ice geometry (e.g., length, slope, width, and bedrock topography), temperature and model integration with time [25]. It is still difficult to predict when the glacial hazards events will happen due to the lack of assessment [23]. Therefore, a detailed analysis of glacial hazard mechanisms and dynamics is of great significance for evaluating future trends [4].

On 17 October 2018, a glacial debris flow/rock fall caused by ice avalanches occurred in the Sedongpu basin of YZR, near the village of Jiala, Milin County, Linzhi city. It blocked the river and drowned several villages; emergency management measures were carried out immediately to manage the barrier lake. This area has experienced many glacier hazards and some preliminary studies have been carried out (e.g., [15,16,19,26]). It provides an opportunity to carry out an in-depth analysis of glacial debris flows and an extensive estimation of the potential hazards associated with glacial changes in the adjacent basins. This paper is structured as follows: the study area is described in Section 2, the data and methods are presented in Section 3, the results and further discussion are presented in Section 4, followed by the conclusions and prospective in Section 5.

## 2. Study Area

The Sedongpu basin (29.81° N, 94.91° E, Figure 1) is located on the western flank of the Jialabailei peak, opposite the Najiabawa peak, which is famous with the "Great Bend" of the YZR Basin as the river flows approximately 180 degrees around Najiabawa [27–29]. The highest elevation in the basin is 7294 m a.s.l. at the main peak of the Jialabailei peak, while the lowest point is 2746 m a.s.l. According to Tong [14], the average slope is approximately 35° and the basin area is ~68 km².

The glaciers present in the Sedongpu basin are mostly monsoonal temperate glaciers and characterised by the presence of debris mantles in the ablation zones that can originate from various sources [30,31]. These glaciers receive snowfall in summer mostly while simultaneously experiencing high rates of ablation, producing a characteristically high ice flow velocity and shear strength [29]. Glacial debris flows, rock falls and landslides occur

frequently on the side of the Jialabailei peak, the western bank of the Great Canyon Region of the YZR [30] (Figure 1).

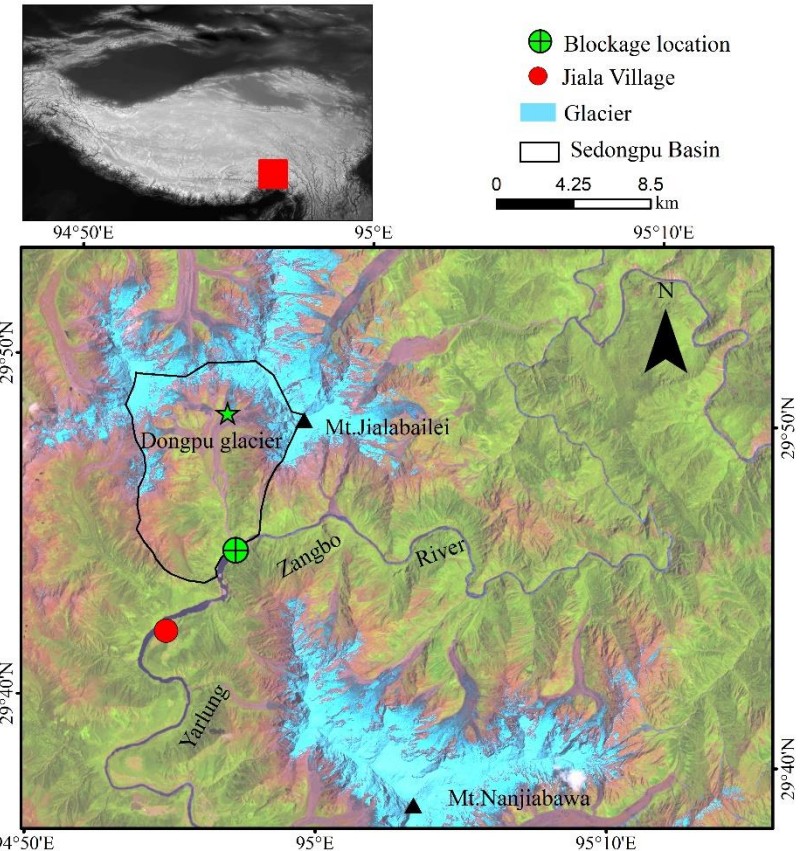

**Figure 1.** Location of the Sedongpu basin, YZR, the image is Landsat 8 in 2015. Note that the Dongpu glacier is the largest glacier in the Sedongpu basin.

## 3. Data and Methods

### 3.1. Data and Processing

A total of two Sentinel-2A images were downloaded from the Sentinel-2 database (http://sentinel-pds.s3-website.eu-central-1.amazonaws.com, accessed on 3 April 2020).The images are available on the USGS (United States Geological Survey, http://glovis.usgs.gov/, accessed on 3 April 2020). A total of 18 Landsat TM/ETM+/OLI/TIRS and Sentinel-2 AMSI were used to analyse the characteristics of glacier change and the mechanism of the glacier hazards in this study (Table 1). For Landsat 8 data, we dilated the cloud mask by 200 m (10 Sentinel-2 pixels) to account for uncertainty in the cloud-masking procedure, and manually checked the images for shadowing elsewhere.

The Shuttle Radar Topography Mission (SRTM, the fourth version) DEM for the topography analysis was from the National Aeronautics and Space Administration (NASA) and the Department of Defense National Imaging and Mapping (NIMA) of the USA [32]. The first Glacier Inventory of China (GIC) data were also consulted to aid in interpreting the glacier outline data. This data was provided by the Cold and Arid Regions Environmental and Engineering Research Institute (CAREERI), Chinese Academy of Sciences. Data on earthquakes in the region of the Sedongpu basin were from the China Seismological Network (http://news.ceic.ac.cn, accessed on 20 March 2020). Meteorological data of Linzhi station which is the nearby national station from the Sedongpu basin were supplied by the Chinese Meteorological Science Data Sharing Service Network (http://cdc.cma.gov.cn, accessed on 20 March 2020).

**Table 1.** Remote sensing images of the Sedongpu glaciers used in this study.

| ID | Receive Date | Sensor | Resolution (m) | Path |
|---|---|---|---|---|
| LC8135039_03920150725 | 2015-07-25 | OLI/TIRS | 30/15 | 135/039 |
| LC5135039_03920040507 | 2004-05-07 | TM | 30 | 135/039 |
| LC5135039_03919960415 | 1996-04-15 | TM | 30 | 135/039 |
| LC5135039_03919870407 | 1987-04-07 | TM | 30 | 135/039 |
| L1C_T46RFT_A004120 | 2017-12-20 | MSI | 10 | T46RFT |
| L1C_T46RFT_A004263 | 2017-12-30 | MSI | 10 | T46RFT |
| L1C_T46RFT_A008982 | 2018-11-25 | MSI | 10 | T46RFT |
| L1C_T46RFT_A009268 | 2018-12-15 | MSI | 10 | T46RFT |
| L1C_T46RFT_A017533 | 2018-10-31 | MSI | 10 | T46RFT |
| L1C_T46RFT_A017962 | 2018-11-30 | MSI | 10 | T46RFT |
| L1C_T46RFU_A004120 | 2017-12-20 | MSI | 10 | T46RFT |
| L1C_T46RFU_A004263 | 2017-12-30 | MSI | 10 | T46RFT |
| L1C_T46RFU_A008982 | 2018-11-25 | MSI | 10 | T46RFT |
| L1C_T46RFU_A009268 | 2018-12-15 | MSI | 10 | T46RFT |
| L1C_T46RFU_A017533 | 2018-10-31 | MSI | 10 | T46RFT |
| L1C_T46RFU_A017533 | 2018-10-31 | MSI | 10 | T46RFT |
| L1C_T46RFU_A017819 | 2018-11-20 | MSI | 10 | T46RFT |
| L1C_T46RFU_A017962 | 2018-11-30 | MSI | 10 | T46RFU |

*3.2. Methods*

The image of Landsat TM/ETM and Sentinel-2 were orthorectified automatically by USGS using the SRTM DEM. In order to highlight the ice, the false color band composition was used for image fusion. Then, the boundary mask was used to extract the study area. The root mean square error (RMSE) of the geometric correction was less than one pixel. All the data are presented in the Universal Transverse Mercator (UTM) coordinate system, and the topographic maps are referenced to the World Geodetic System 1984 (WGS84, UTM zone 46N). Debris hampers the mapping of the actual ice snout by means of space-borne imagery due to the spectral similarity to the surrounding bedrock. In our study, a preliminary glacier boundary was generated automatically using the band ratio method for clean ice [33]. Principal component analysis (PCA) was performed to determine the spectral differences between bare rock and debris-covered glaciers, in the software ENVI with bands five and seven centred and standardised.

Elmer/Ice dynamical model was chosen in this study, for it has proved to be a valid tool to simulate glacial surface velocity, especially in mountain glaciers [34]. Considering the classic theory in glacier physics (e.g., [35]) and field experience, the mesh size was set to 80 m and the mean ice thicknesses of two scenarios were given at approximately 28 m (S1) and 100 m (S2), respectively. The initial parameters are given in Table 2. The two important parameters, the Glen enhancement factor (*E*) and the basal friction parameter (*β*), were set to 2.0 and 0.06, respectively, in both simulations.

**Table 2.** The parameter of the ice-flow model.

| Symbol | Description | Value and Unit |
|---|---|---|
| $\rho$ | Ice density | $910 \ kg \ m^{-3}$ |
| $g$ | Gravitational acceleration | $9.81 \ kg \ s^{-2}$ |
| $n$ | Glen exponent | 3 |
| $A_0$ | Rate factor | |
| | When T $\leq -10$ °C | $2.89 \times 10^{-13} \ s^{-1} \ Pa^{-3}$ |
| | When T $> -10$ °C | $2.43 \times 10^{-13} \ s^{-1} \ Pa^{-3}$ |
| $Q$ | Creep activation energy | |
| | When T $\leq -10$ °C | $60 \ kJ \ mol^{-1}$ |
| | When T $> -10$ °C | $115 \ kJ \ mol^{-1}$ |
| $R$ | Gas constant | $8.31 \ J \ kg^{-1} \ K^{-1}$ |

In addition, a detailed database of historical glacier hazards has been compiled for the Sedongpu basin and regions nearby, based on previous scientific literature, reports,

media news, and assessments of observations (e.g., [36,37]). These data were re-analysed and arranged to analyse the temporal and spatial patterns of glacier hazards in this region.

*3.3. Uncertainties*

Errors in the glacier boundaries from remote sensing images were controlled by the image resolution and co-registration error. The uncertainty can be calculated by the following formulae [38,39]:

$$U_T = \sqrt{\sum \lambda^2} + \sqrt{\sum \varepsilon^2} \tag{1}$$

$$U_A = 2U_T\sqrt{\sum \lambda^2} + \sqrt{\sum \varepsilon^2} \tag{2}$$

where $U_T$ is the uncertainty of glacier length; $\lambda$ is the image resolution; $\varepsilon$ is the co-registration error between each image and the topographic map; and $U_A$ is the uncertainty in the glacier area. The uncertainty in glacier-change area measurements is calculated with an accuracy of $\pm 0.003$ km$^2$ using TM or ETM+ imagery and $\pm 0.002$ km$^2$ for Sentinel-2A MSI.

## 4. Results and Discussion

*4.1. Blockage Event Process*

The event occurred on the side of the Jialabailei peak, the left bank of the YZR Great Canyon Region, on 17 October 2018. Figure 2a shows a large amount of loose moraine debris flow on the steep slope near the glacier, which is an important precondition for the formation of a glacier disaster. When an ice avalanche occurred at the glacier upstream, the fast-moving ice mass intensely scraped and eroded the debris in the Dongpu glacier downstream, and the debris accumulated along the valley and formed a dam. Then, the water level kept rising (Figure 2b), threatening the Jala village and electrical facilities (Figure 2c,d).

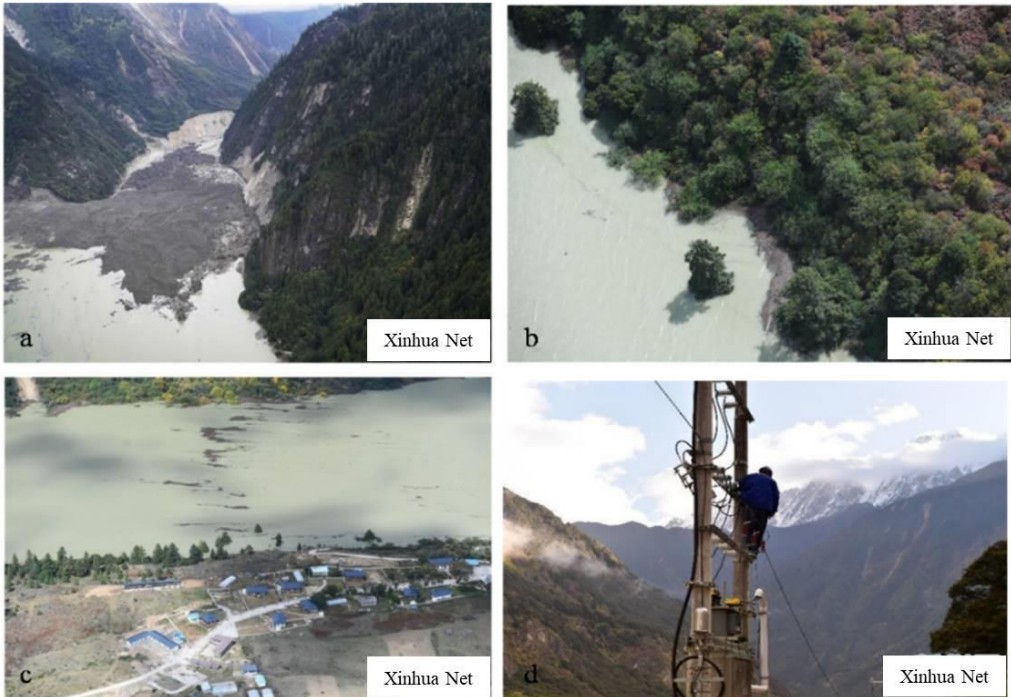

**Figure 2.** Photographs of the conditions following glacial debris flow event: (**a**) the dammed lake and the barrier body; (**b**) the rising water level of the barrier lake; (**c**) flooding in the Jala village; (**d**) the electric power department cut the power to prevent accidents. The Chinese blue writing means the source of the photographs is from Xinhua Net (http://www.xinhuanet.com/photo/2018-10/18/c_129974593.htm, accessed on 10 March 2019).

### 4.2. Historical Glacier Hazards

Due to the high logistical requirements and difficult working conditions, in situ studies on glacier hazards are rare, and most are supported by remote sensing techniques (e.g., [30]). Without the availability of other options, we also used a variety of high- to medium-resolution satellite images from 2017 to 2018 to identify the glacial debris flow events here (Figure 3).

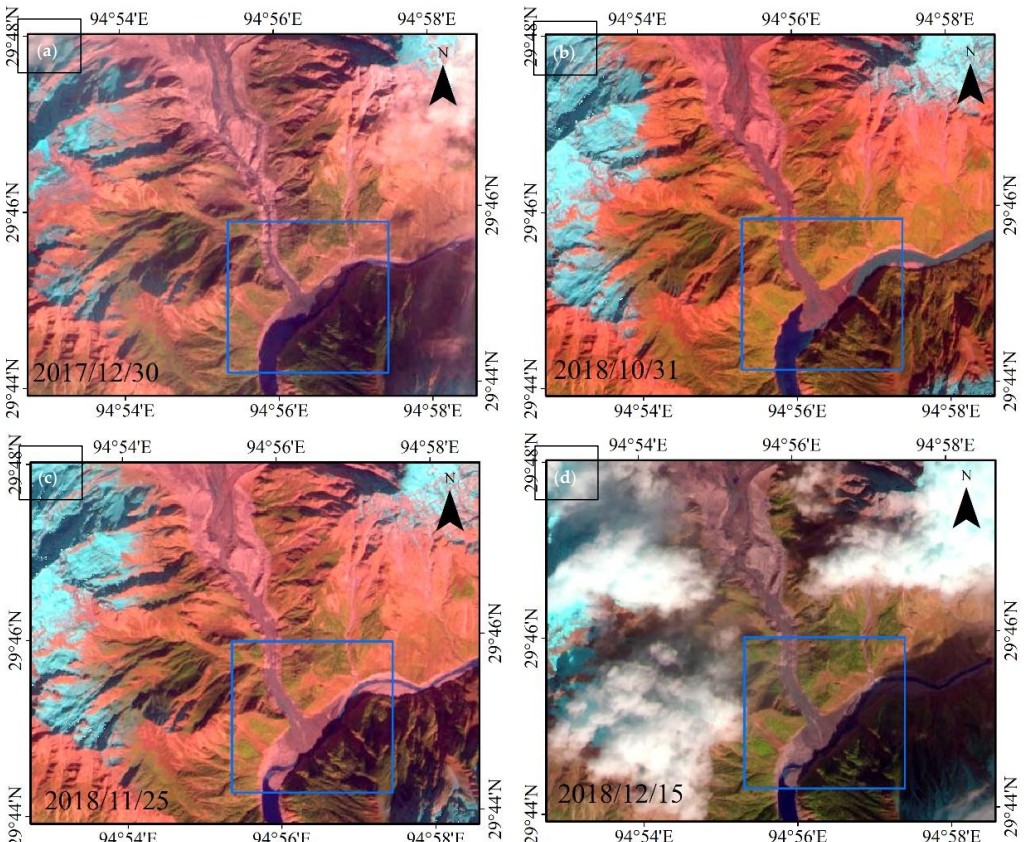

**Figure 3.** Examples of river-blocking events on images of Sentinel-2 in Yarlung Zangbo River: (**a**) 30 December 2017; (**b**) 31 October 2018; (**c**) 25 November 2018; (**d**) 15 December 2018; the blue box is the region blocking the river.

In conjunction with the historical database of glacier hazards in the Sedongpu basin (Table 3), many villages have been threatened by glacier hazards, including the villages of Zhibai, Gega, Gu, Pailong, Songrao, Bitong, Baka, Layue, Suotong, Galang, and Jiala, most of which are on the left bank of the Yarlung Zangbo River (Table 3 and Figure 4). The ice avalanches, glacial lake outburst floods, glacier surges and glacial debris flows have caused severe damage and shaped local living, but glacial debris flow is the dominant hazard (e.g., [40–42]. Based on the location frequency of the occurrence, we found that the Nanjiabawa Peak was the most dangerous place. In addition, these cryosphere hazards typically occurred between May and September, coinciding with the period of intense ablation of monsoonal temperate glaciers, as has been pointed out [31].

**Table 3.** Historical glacier hazard events in the Sedongpu basin.

| Date | Location | Hazard Type | Reference | Village Affected |
|---|---|---|---|---|
| 1950, 1968, 1984 | Zenong glacier | Glacier debris flows<br>Glacier surge | [40]<br>[43]<br>[44] | Zhibai village<br>Gega village |
| 1953, 1972, 2005 | Guxiang Valley | Glacier debris flows<br>GLOF | [45]<br>[36]<br>[37] | Gu village |
| 1983–1985 | Peilong Valley | Glacier debris flows<br>GLOF,<br>ice avalanches | [42]<br>[41]<br>[46] | Pailong village |
| 2007 | Tianmo Valley | Glacier debris flows | [47] | Songrao village |
| 2007 | Bitong Valley | Glacier debris flows | [47] | Bitong village |
| 2007 | Baka Valley | Glacier debris flows | [47] | Baka village |
| Unknown | Layue | Glacier debris flows | [26] | Layue village |
| Unknown | Suotong Valley | Glacier debris flows | [48] | Suotong village |
| Unknown | Gelang Valley | Glacier debris flows | [48] | Galang village |
| 2017, 2018 | Sedongpu basin | Glacier debris flows<br>Glacier avalanches | This study | Jiala village |

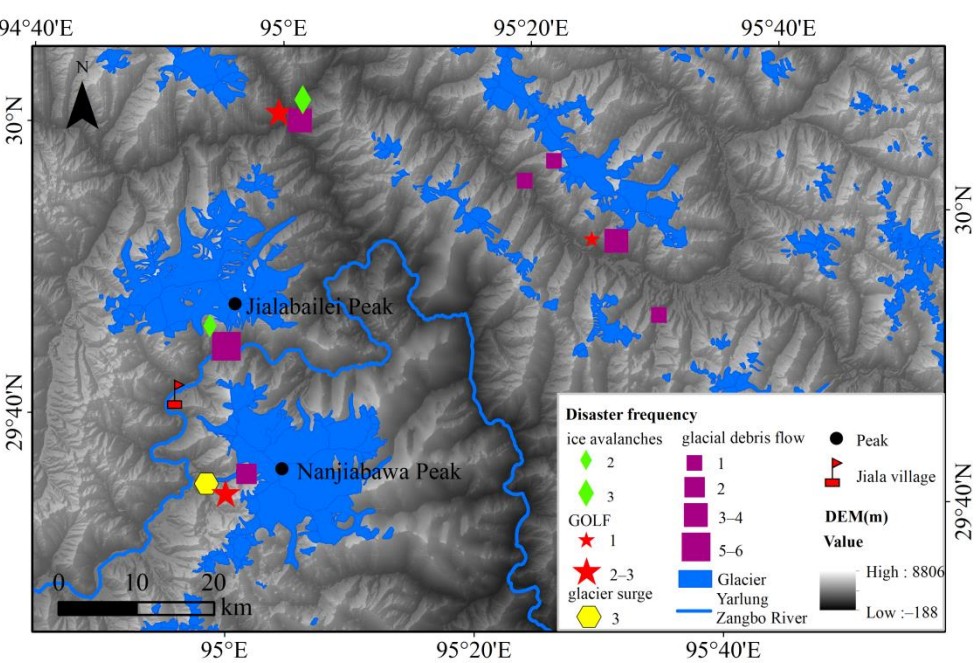

**Figure 4.** Frequency distributions of ice avalanches, glacial lake outburst floods (GOLF), glacier surges and glacial debris flows during the 1950s~2018 period.

### *4.3. Diagnostic of Influencing Factors*

Many previous researchers have studied the factors influencing glacier hazards, including weather, hydrological conditions, active tectonics and topography (e.g., [14,26]). In this paper, we analysed the effects of climate change, topography, precipitation, earthquake, debris cover and the internal glacial conditions.

#### 4.3.1. Climate Change

Based on the meteorological records of the Linzhi station (Figure 5a,b), the average annual temperature and precipitation were 8.8 °C and 680.0 mm, respectively, during the 1960–2017 period. The annual temperature fluctuated from 7.4 °C to 10.3 °C, while annual precipitation varied from 452.44 mm to 985 mm during the 1960–2017 period at the Linzhi station. The average summer temperature is 13.5 °C, which is 3.2 °C higher than the annual

average, while the monthly temperature is at its highest in July with a value of 16.2 °C. The warming rate was greater than 0.40 °C/10a (statistically significant at the 0.01 level) during the 1960–2017 period (Figure 5a), which is twice the global average [49]. Therefore, with such regional climate changes in recent decades in Sedongpu basin, these temperate glaciers were experiencing substantial mass loss and large-scale shrinkage [22,29]. Furthermore, the succession of warm summers could potentially warm the frozen ice-bedrock interface, resulting in a reduction of the basal support and produce melt water due to pressure and basal warming [10], indicating glacier instability in the Sedongpu basin as a consequence of regional climate change.

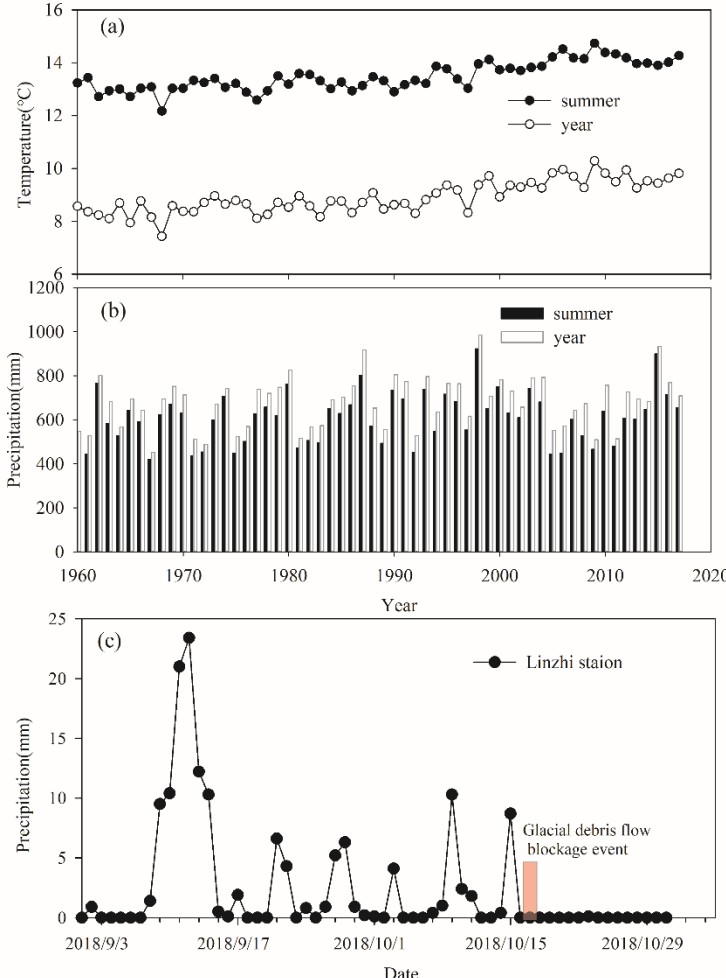

**Figure 5.** The variations in annual temperature and summer temperature (**a**); annual precipitation and summer precipitation (**b**); and daily precipitation before and after the glacial/rock debris flow event (**c**). All data are sourced from the Linzhi meteorological station.

During the 1960–2017 period, the annual average precipitation at the Linzhi station reached 680 mm/a (Figure 5b), with the rate increasing by 17.3 mm/10a during the 1960–2017 period. The precipitation was mainly concentrated in June–September, accounting for almost 80% of the annual precipitation. The monthly precipitation was at its highest in July with a value of 130.4 mm. It is worth noting that, 5 days before the first avalanche (12 October 2018), there was heavy rainfall with daily precipitation of 9.3 mm (Figure 5c). Observations and modelling have proven that rainfall has accelerated the warming and melting of snow/ice in some Tibetan Plateau cryospheric regions [50]. If rain occurs in the ablation area, it may flow to the glacier bottom through surface melt ponds/conduits and then accelerate the glacier ice flow [31].

### 4.3.2. Topography

The elevation difference from the top to the bottom (A to B in Figure 6) of the glaciers in the Sedongpu basin is about 3500 m, and the average slope is as high as 48°. These conditions are favourable not only for abundant precipitation from orographic-convective clouds, but also for rapid ice flow. In the upper part of the Dongpu glacier (Figure 6c,d), the slope is extremely steep, leading to a high instability of ice and snow, and will result in further snow/ice avalanches and collapses [51].

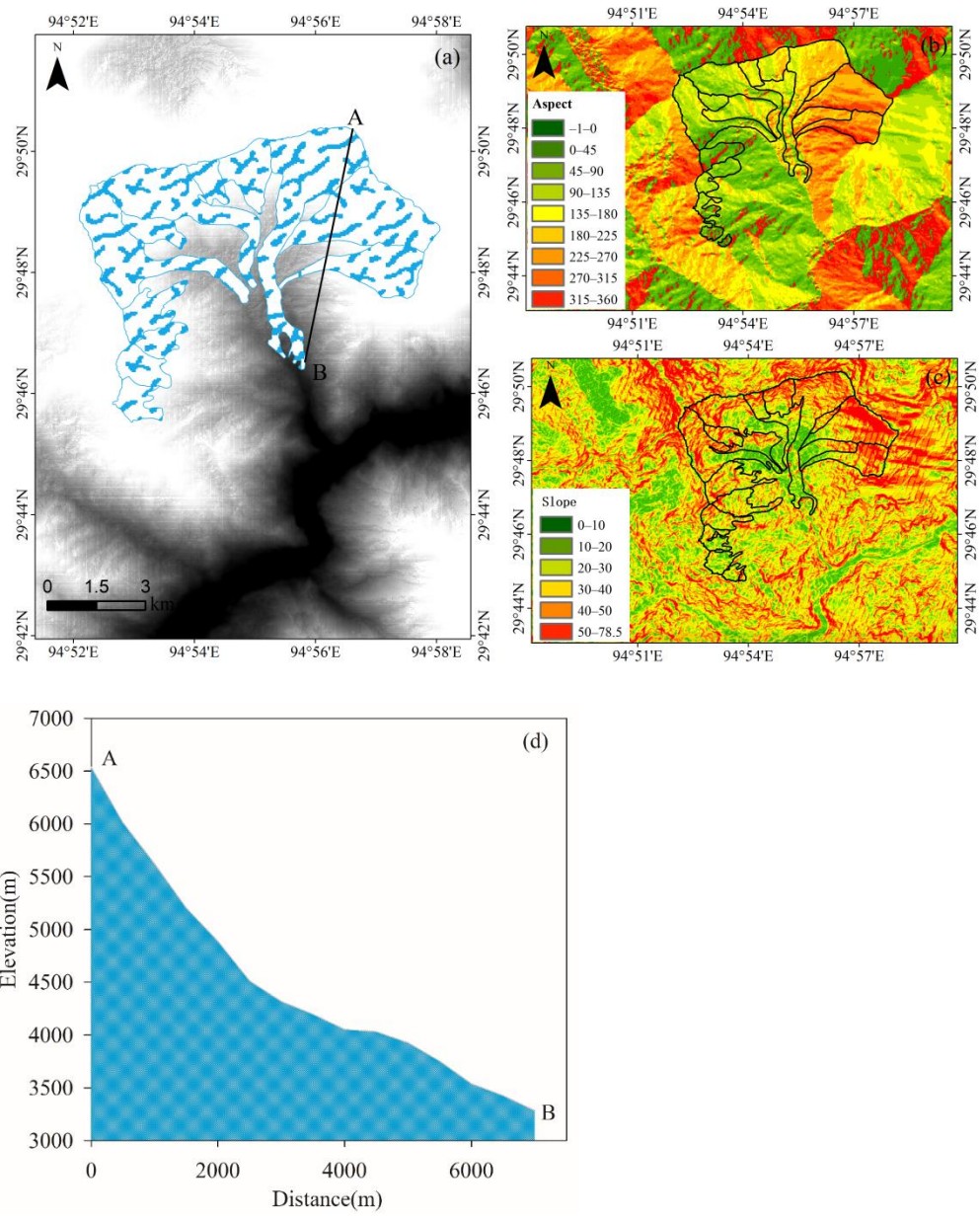

**Figure 6.** Glacier terrain based on SRTM in the Dongpu glacier, (**a**) points A to B on the Dongpu glacier as an example of the measurement of elevation differences. (**b**) The aspect of the glaciers in Sedongpu basin within the black line. (**c**) The slope of the glaciers in Sedongpu basin within the black line. (**d**) The elevation difference between points A and B on the Dongpu glacier.

Compared with the upstream of the Sedongpu basin, the downstream valley is relatively narrow. Glacial debris and moraine deposits would accumulate along with melting and small-scale snow/ice avalanches. Consequently, the destructive capacity of the next large-scale ice avalanche increases. In some cases, large-scale avalanches can mobilize the debris and induce the initiation of large-scale landslide debris flow.

### 4.3.3. Glacier Dynamics

In the S2 case, the maximum ice flow velocity is 705.0 m/a. S1 is much slower, 72.8 m/a, but it is still a very rapid speed compared with the other glaciers (Figure 7). An et al. [51] found that the glacier speed in the Sedongpu basin increased sharply in August–September according to the SAR slant range direction and azimuth direction from the Sentinel-1A images. During the period April–May to August–September in 2018, the speed increased rapidly from around 15 cm/d to 40 cm/d. The Elmer/Ice model simulated results with 19 cm/d in the case region of our study agreed well with the analysis derived from SAR technology [51]. The location with the highest ice flow coincided with the ice avalanche initial point. It indicated that the basin cannot hold too much ice in such a steep topography. Plus the heavy precipitation here, the volume response time of the glacier would be very short, at least shorter than most of the glaciers.

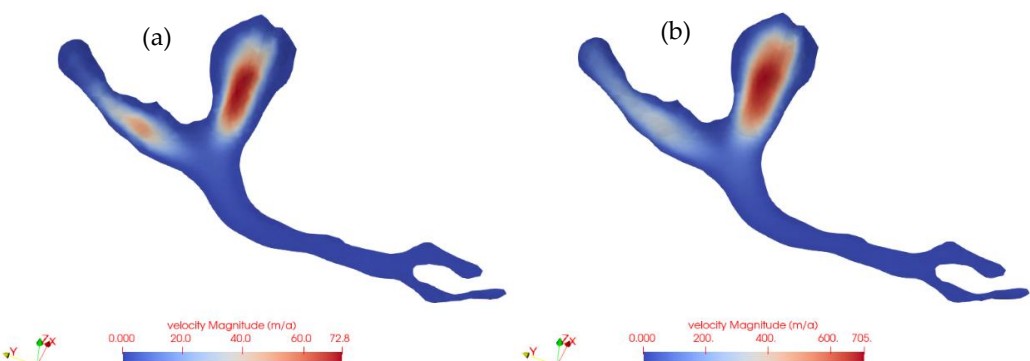

**Figure 7.** Simulated glacial ice flow velocities using Elmer/Ice with parameters in Table 3. (**a**) Left pane shows the scenario with mean ice thickness of 28 m, which has a maximum ice flow velocity of 72.8 m/a; (**b**) Right pane shows the scenario with mean ice thickness of 100 m, which has a maximum ice flow velocity of 705 m/a.

### 4.3.4. Earthquakes

Earthquake is considered as an immediate triggering factor and responsible for the massive accumulation of loose mass [52]. There were many great earthquakes along YZR region, such as the Ms 7.3 earthquake on 14 August 1932, and the Ms 8.6 earthquake on 15 August 1950 [15]. Based on data and assessment of the Chinese Seismological Network, seismicity has been active in the Jialabailei peak region since 2010. In the recent 3 years (2017–2019), there were as many as 18 earthquakes, of which 11 had magnitudes of Ms 3–3.9, 5 had magnitudes of Ms 4–4.9, one had a magnitude of Ms 5.9 and one had a magnitude of Ms 6.9 (Figure 8). Compared with the southern bank of the river, the northern bank has more earthquakes and most of them happen near the Jialabailei peak. After the Linzhi 6.9 Ms earthquake, the deformation associated with glacial debris flows in the Sedongpu basin was further aggravated, and the central deformation was significantly higher in this region than in other regions [14,53], indicating that the earthquake had a clearly destructive effect and direct influence on the glaciers. One piece of evidence is that the Dongpu glacier surface velocity accelerated after the Linzhi 6.9 Ms earthquake, as shown by remote sensing images [15]. Another study also proposed that an earthquake in Nagqu (about 10 min before the landslide event) also played an important role in triggering the Sedongpu debris flow blockage event [9]. However, the Linzhi earthquake was the decisive factor in this event, with the significant acceleration of the glacier surface velocity after the Linzhi earthquake, while the Nagqu earthquake magnitude was 4.1 Ms and the long distance weakened the impact.

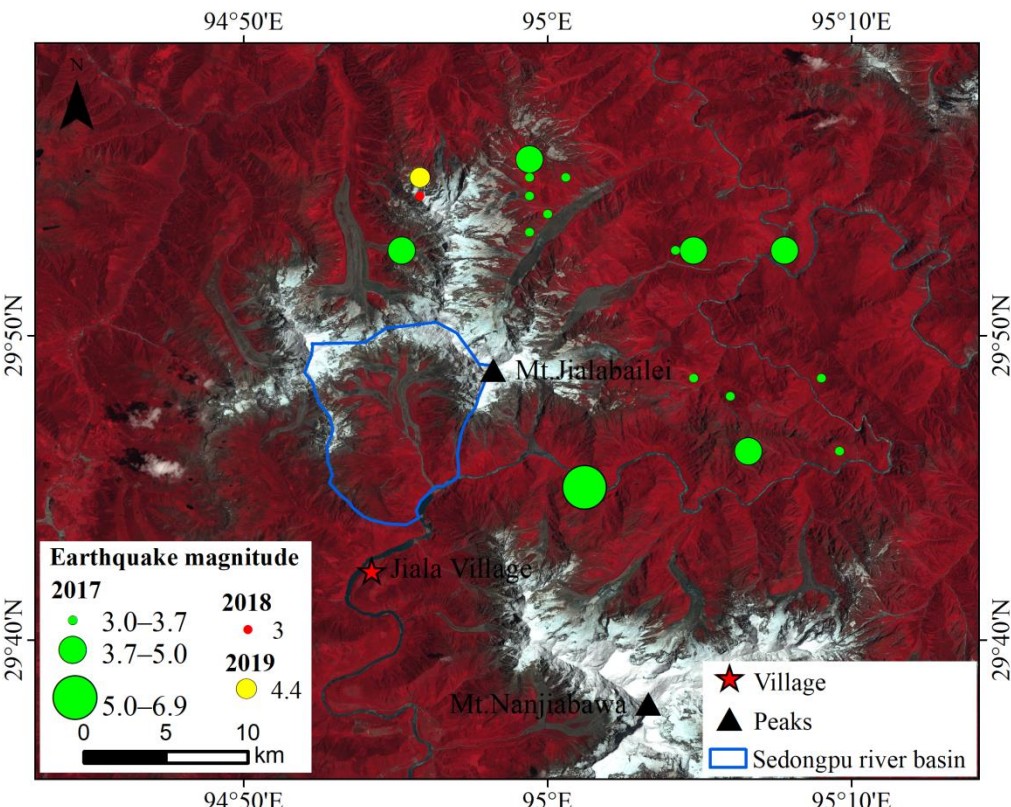

**Figure 8.** Distribution of earthquakes during the 2017–2019 period in Sedongpu basin and surrounding regions; the background image is Sentinel-2 in 2018.

### 4.3.5. Characteristics of Debris in the Glacier Ablation Area

The debris cover is well developed in Sedongpu basin, based on the image, especially on the large valley glaciers, where a large area of the ablation zone is covered by thick debris that extends to relatively high elevations [54]. It is estimated from the remote sensing images that the debris coverage occupies half of the area of the Dongpu glacier (Figure 1). Due to glacial erosion and rockslides, these debris-covered areas have been supplied with a large amount of mass for the glacial debris flow to occur in the basin. This is consistent with the opinion of Tong et al. [14]. Studies also have shown that ablation areas covered with debris retreated at a lower rate than many other non-debris-covered glaciers on the TP [55,56]. If the terminal region is covered with debris, the ice volume loss in the upper ablation area (at higher altitudes) can be greater in magnitude than the terminus [56]. Compared to clean ice, a 1 cm thick debris cover can reduce the energy flux available for melting by 33% if the debris is dry and by 11% if the debris is wet [57]. However, the expansion of the debris area in the Sedongpu basin also provides more mass and enhances the destructiveness of debris flows.

## 5. Conclusions and Prospective

In summary, the river blockage incident on 17 October 2018 was induced by a large-scale ice avalanche from the Dongpu glacier in the Sedongpu basin. The elevation difference of the Dongpu glacier is about 3500 m and the average slope is as high as 48°, leading to a high instability of ice. Based on the meteorological records of the Linzhi station, the warming rate was greater than 0.40 °C/10a ($p > 0.01$) during the period 1960–2017. It is worth noting that 5 days before the blockage event, there was heavy rainfall with a daily value of 9.3 mm. Short-term heavy rainfall is regarded as a triggering factor for glacier debris flows. In addition, the Dongpu glacier surface velocity accelerated after the Linzhi 6.9 Ms earthquake, as shown by remote sensing images.

Nevertheless, this study was mainly carried out with some qualitative analysis, similar to the other Himalayan glaciers. An insufficiency of field data (mass balance, ice thickness, velocity, etc.) make it difficult to develop a coherent picture. The suggestions we propose may have many deficiencies. Therefore, countries around the Tibetan Plateau should increase their investment in the monitoring, research and early warning of glacier hazards.

**Author Contributions:** M.D. and B.H. provided the topic and idea, B.H., M.D. and W.S. coordinated the study and carried out the analysis; S.A. carried out the ice-flow model analysis; M.D. and B.H. drafted the paper, S.A., Y.W. and J.G. edited the paper, J.G. revised the paper. All authors have read and agreed to the published version of the manuscript.

**Funding:** This research was funded by the National Natural Science Foundation of China (41690143 and 41671058) and the Basic Foundation of the Chinese Academy of Meteorological Science (2019Z008).

**Data Availability Statement:** Remote sensing data used in this paper is available at http://sentinel-pds.s3-website.eu-central-1.amazonaws.com. The Glacier Inventory of China data can be downloaded from http://westdc.westgis.ac.cn/zh-hans/data/. Meteorological data is available at http://cdc.cma.gov.cn. All accessed on 1 June 2022.

**Conflicts of Interest:** The authors declare no conflict of interest.

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
