# Peer review of "Glacial Debris Flow Blockage Event (2018) in the Sedongpu Basin of the Yarlung Zangbo River, China: Occurrence Factors and Its Implications"

_land, doi:10.3390/land11081217_

Round 1

Reviewer 1 Report

Manuscript ID: land-1777454

Title: Glacial debris flow blockage event (2018) in the Sedongpu basin of the Yarlung Zangbo River, China: Occurrence factors and its implications

Authors: Baojuan Huai, Minghu Ding, Songtao Ai, Weijun Sun, and Yetang Wang

In this work the glacial debris flow blockage event occurred on October 17th, 2018, in the Sedongpu basin of the Yarlung Zangbo River (Tibet, China) is taken by authors as an example to analyse the occurrence and development of glacier disasters in this region. Moreover, authors demonstrate that the topography and climate background determine the hazard happens periodically, and they are fundamental for the weak vulnerability of this area.

In this form this manuscript cannot be accepted for publication on MDPI-Land, in fact some changes are necessary to fit the high-quality standards of this journal.

As general remarks:

1)              Authors should format the text of this manuscript using the template provided by the journal at this link (Word version):

https://www.mdpi.com/files/word-templates/land-template.dot, in particular for the format of the references that are to be indicated as numbers [1,2] ….. Etc, etc.

2)              In the section Introduction, I suggest authors to add some sentences describing the contents of the following section of the manuscript. In practice how the manuscript is organised.   

As particular indications:

1)              At line 9 of the section Abstract: in the sentence “The results indicated that the glacier disasters may seriously affected”, I think that the word affect in place of affected should be used.

2)              In Figure 1 at page 3, Authors should add the Orientation bar (North, South, East, West) as in Figure 3.

3)              At the beginning of page 4, after table 1, authors should indicate the version of SRTM Dem used and add corresponding references.

4)              At line 17 of the section 3.2 Methods, I think that authors intended “field experience” and not “filed experience” as indicted in the text.

5)              At page 5 – In the Figure 2 - I suggest authors to translate the Chinese blue writes in the figure or to remove them if are not necessary.

6)              At page 8 – In the Figure 5 - (a) and (b) letters are not present in the Figure, the (c) letter is repeated and the (d) is lacking (recalled also at the end of the page).

7)              At page 10 – lines from 24 to 29 – At the end of this page, authors should compare the computed velocities with SAR offset tracking methodology. Adding some references.

8)              At page 10, first line of the section 4.3.4 Earthquakes, I think that landslide can occur after earthquake depending also on the location and magnitude of the events, authors should add some references.

9)              In Figure 8, page 11, I suggest authors to add the orientation bar as done in Figure 3, for example, to indicate the direction of the North.

Based on the indications above I recommend a major revision of this manuscript.

Reviewer 2 Report

Dear authors, also this is definitely an important study, the structure of the manuscript including the description of many parts and results should be improved. Please see the major comments below and more details in the attached pdf document.

1.     Abstract could be improved. There are some issues which are not clear, see the pdf file. Also, I miss in the beginning of the abstract information on the used methods, thus it is not clear what do you mean by synthesis study at the end, what kind of data did you synthesise.

2.     First two highlights should be improved because they are not understandable without context.

3.     Introduction: should be improved, there are some English issues, and some sentences need to be clarified and rewritten. Also, the introduction reads more like a study area and previous investigations, some parts could be removed to the Study area. Then, the introduction could be extended describing the glacier disasters, processes and mechanisms in more detail.

4.     Study area is too short and could be extended giving more details on the glacier disasters in the region, this may be done by moving some parts from the Introduction.

5.     Methods: should be improved. Many important details are lacking, see the comments in the pdf file.

6.     4. Results and discussion: this is the most important chapter of the manuscript and it needs complete rewriting. First of all, the structure is not suitable. Authors frequently mix the description of the study area, methods, discussion, and results. Usually, subchapters start with some speculations and discussion, which then are followed by some results. From such a structure I actually did not get what are the main results. You should write the results and interpretation first and separately. Mainly this chapter is based on discussion, and some parts of it are out of context and not relevant to the findings of this study, which are described weakly. See more comments in the pdf file.

7.     Conclusions: These sentences with references should not be included in conclusions: “The cryosphere is considered as an amplification of climate change, especially the
Arctic, west Antarctica, and the Tibetan Plateau (Xiao et al., 2015; Qin et al., 2018). Under global warming, the increase in air temperature at high-elevation areas is obviously greater than the global mean (An et al., 2017), accompanied with shrinking glaciers and intensified glacial activities all over the world (Pritchard, 2019; Milillo et al., 2019; Maurer et al., 2019)”.

And these as well should be removed: “Except Sedongpu basin, the frequency and scale of debris flows in Tibetan Plateau are likely to increase sharply in the future. Some of severe cases may block the large rivers and threaten people downstream, for example, the events on the Jinshajiang River on October 11th 2018 and on the Alakananda/Dhauliganga rivers in the state of Uttarakhand, India, on February 7th 2021”.

Also, the last paragraph of the conclusions refers to literature without references, it should be removed as well. In the end, there are basically no conclusions related to the findings of this study. Please write them.

Round 2

Reviewer 1 Report

Dear Authors of manuscript land-1777454,

I am happy to inform you that the reviewing process of this paper has been completed successfully, and the manuscript is now ready for publication on prestigious journal Land of MDPI.

Author Response

We thank the reviewer for the comments, which have improved the paper.

Reviewer 2 Report

Dear authors, thank you for submitting the revised manuscript, you have made some changes but many issues are still not addressed and there is no response to some comments, so I repeat them again and encourage you to improve the manuscript. Some minor comments are in the pdf file, by yellow colour I have marked words, which are not correct from the English perspective.

  Abstract is exactly the same as previously. No suggested changes were made.

2.     Introduction: The authors claim that they have removed some parts to the “Study area” part and have re-organized the “Introduction” part and extended describing the glacier disasters, processes and mechanisms in more detail but they have not done anything of the mentioned as suggested in the previous review.

3.      Methods: there is no answer to the previous question “where this station is located? Can you show it on the map?”

Line 108: “Preprocessing of image data included accurate geometric correction and image fusion” -  there still is no detailed information on the preprocessing steps of images.

Line 110: mention the used zone of the UTM coordinate system.

Instead of providing the necessary details on the modelling, the authors have moved the paragraph from the Methods section related to ice flow modelling to Results.  

6.     Results and discussion: authors have deleted some not relevant parts but the main issues are not addressed. as I wrote previously: this chapter needs complete rewriting. First of all, the structure is not suitable. Authors frequently mix the description of study area, methods, discussion, and results. Usually, subchapters start with some speculations and discussion, which then are followed by some results. From such a structure I actually did not get what are the main results. You should write the results and interpretation first and separately.

No answer to the previous remark - Also, can you put the names of some largest villages in the map? and/or maybe use some background - satellite image or elevation model?

4.3.1. Climate change: there was a major comment, which is not addressed at all: “this paragraph sounds too speculative without data and or references”.

Line 201-202: which regions?

Previous comment: “you should start this paragraph by this last sentence and then describe more your results not begin with some speculations.” – no more results are provided.

Line 209: no answer to the previous comment: do you have any data on ice velocity or at least references to it, without data, it is just speculation?

Fig.6.: no response to the previous remark: can you denote the area covered in b and c in fig. a?

Lines 232-238: the authors have moved the paragraph related to modelling from the Methods here but this is the Result chapter, it should be moved back. Also, the answer is required to the previous comment: why did you choose exactly such glacier thickness?.

Lines 241-242: please explain what exactly agrees well with the analyses derived from SBAS-InSAR.

Line 277: which region?

Fig.8. No response to the previous comment: please show your study area here and the most prominent peaks and or villages.

Chapter 4.3.5. Still no improvements: what are your results regarding the debris cover? This reads more like a description of the study area, please give more details on the debris cover and its influences.

Conclusions: The authors have deleted the previous middle paragraph of conclusions leaving the rest the same. The authors have not written any new conclusion as suggested previously. The conclusions mention very general issues but not the results of this study thus conclusions still need to be rewritten.
